# Pharmacomicrobiomics and Drug–Infection Interactions: The Impact of Commensal, Symbiotic and Pathogenic Microorganisms on a Host Response to Drug Therapy

**DOI:** 10.3390/ijms242317100

**Published:** 2023-12-04

**Authors:** Norma Torres-Carrillo, Erika Martínez-López, Nora Magdalena Torres-Carrillo, Andres López-Quintero, José Miguel Moreno-Ortiz, Anahí González-Mercado, Itzae Adonai Gutiérrez-Hurtado

**Affiliations:** 1Departamento de Microbiología y Patología, Centro Universitario de Ciencias de la Salud, Universidad de Guadalajara, Guadalajara 44340, Mexico; norma.tcarrillo@academicos.udg.mx (N.T.-C.); nora.torres@academicos.udg.mx (N.M.T.-C.); 2Instituto de Nutrigenética y Nutrigenómica Traslacional, Centro Universitario de Ciencias de la Salud, Universidad de Guadalajara, Guadalajara 44340, Mexico; erika.martinez@academicos.udg.mx; 3Departamento de Biología Molecular y Genómica, Centro Universitario de Ciencias de la Salud, Universidad de Guadalajara, Guadalajara 44340, Mexico; andres.lopezq@academicos.udg.mx; 4Instituto de Genética Humana “Dr. Enrique Corona Rivera”, Centro Universitario de Ciencias de la Salud, Universidad de Guadalajara, Guadalajara 44340, Mexico; miguel.moreno@academicos.udg.mx (J.M.M.-O.); anahi.gonzalez@academicos.udg.mx (A.G.-M.)

**Keywords:** pharmacomicrobiomics, toxicomicrobiomics, pharmacoecology, drug–infection interaction, microbiome

## Abstract

Microorganisms have a close relationship with humans, whether it is commensal, symbiotic, or pathogenic. Recently, it has been documented that microorganisms may influence the response to drug therapy. Pharmacomicrobiomics is an emerging field that focuses on the study of how variations in the microbiome affect the disposition, action, and toxicity of drugs. Two additional sciences have been added to complement pharmacomicrobiomics, namely toxicomicrobiomics, which explores how the microbiome influences drug metabolism and toxicity, and pharmacoecology, which refers to modifications in the microbiome as a result of drug administration. In this context, we introduce the concept of “drug-infection interaction” to describe the influence of pathogenic microorganisms on drug response. This review analyzes the current state of knowledge regarding the relevance of microorganisms in the host’s response to drugs. It also highlights promising areas for future research and proposes the term “drug-infection interaction” as an extension of pharmacomicrobiomics.

## 1. Introduction

A pharmacological interaction is a situation in which the activity of a medication is affected because it is administered simultaneously with another drug, with certain food, or due to extrinsic or intrinsic factors. Pharmacological interactions can lead to the development of medical complications, mainly because they can reduce therapeutic effectiveness or increase the toxicity of pharmacological treatment. According to the World Health Organization, at least 60% of adverse drug reactions could be avoided. In this context, drug interactions stand out as the primary cause of these adverse reactions [1,2].

There are several ways to classify drug interactions; however, the most common approach is to classify them according to their mechanism of action: into pharmacokinetic or pharmacodynamic interactions. The former refers to those that affect how a drug is absorbed, distributed, metabolized, or eliminated in the body, while the latter refers to how drugs interact directly with biological systems to produce therapeutic or side effects [1,2,3,4].

A pharmacological interaction is considered clinically relevant when it significantly affects the therapeutic efficacy or safety of a medication to the extent that a dose adjustment or a complete treatment change is required. Different regulatory agencies, such as the U.S. Food and Drug Administration (FDA), the European Medicines Agency (EMA), and the Japan Pharmaceutical and Medical Device Agency (PMDA), have developed guidelines for determining the clinical relevance of pharmacological interactions. In general, these three agencies attach great importance to cytochrome P450 (CYP) enzymes, and drug transport proteins in drug interactions [3,5].

In current clinical practice, five main types of pharmacological interactions are primarily considered: drug–drug interactions, which refer to how one drug can modify the response of another drug; drug–disease interactions, which relate to how a health condition can modify the response to a drug; drug–food interactions, which address the influence of food on drug response; pharmacogenetics, which studies how a person’s genetic composition modifies the response to a drug; and drug–substance interactions, which include interactions between drugs, medicinal herbs, supplements, and alcohol [2,3,6].

In the scientific community, the study of how microorganisms influence drug response has been of increasing interest in recent years. Since 2010, when the term “pharmacomicrobiomics” was coined to describe the effects of the microbiome on drug absorption, activity and toxicity, numerous research studies have been conducted on this topic [7,8,9].

In the context of pharmacomicrobiomics, terms like “toxicomicrobiomics” and “pharmacoecology” have emerged. The most recent term is “pharmacoecology”, which was proposed in May 2023, with the purpose of complementing the concept of pharmacomicrobiomics [10]. Initially, this term was proposed in 2008 by C. Flexner to refer to environmental influences on drug disposition and response, although, with this definition, it was only used in three publications [11,12,13]. Due to the limited use of the term and in an effort to provide a more precise definition, Alya Heirali et al. proposed that the term “pharmacoecology” be employed to conceptualize modifications in microbial taxa or specific functions of the microbiome as a result of the administration of a microbicidal or pro-microbial drug [10,14]. On the other hand, “toxicomicrobiomics” specifically refers to the study of how variations in the microbiome affect the metabolism and modify the toxicity of xenobiotics, including drugs [15,16].

In general, pharmacomicrobiomics focuses on interactions between drugs and the “microbiome”. The microbiome, and even dysbiosis, specifically refer to symbiotic or commensal microorganisms. It is worth mentioning that, in this context, pharmacomicrobiomics does not provide a specific definition for interactions between drugs and pathogenic microorganisms, because an infection is characterized by the invasion and proliferation of pathogenic microorganisms in body tissues [9,17,18]. As far as we know, there is no specific term like “pharmaco-infection interaction” in the current literature. However, in this document, we will use this term to refer to the effect of pathogenic microorganisms and their influence on drug pharmacokinetics and/or pharmacodynamics. In Figure 1, the distinction between pharmacomicrobiomics, pharmacoecology, toxicomicrobiomics, and drug–infection interaction is made clear.

The aim of this study was to conduct a review of the available evidence of the impact of microorganisms, whether pathogenic, commensal, or symbiotic, on the response to drug treatment. In addition, we propose to incorporate the term “drug-infection interaction” to distinguish interactions generated by the microbiome and pathogenic microorganisms in response to a drug.

## 2. Effect of the Microbiome on Drug Response: “Pharmacomicrobiomics”

To easily grasp the term “pharmacomicrobiomics”, it is essential to begin by defining and clearly distinguishing between the concepts of microbiota and microbiome. Although they are sometimes used interchangeably, they have significant differences. Microbiota refers to the organisms that maintain a symbiotic relationship with humans, while the microbiome encompasses both these organisms and their genetic composition, as well as their interaction with the host’s genome [8,17].

Pharmacomicrobiomics is defined as the study of the effect of variations in the microbiome on the disposition, action, and toxicity of drugs. Although most of the research related to pharmacomicrobiomics is centered on the intestinal microbiota, it is important to note that there are five specific regions in the human body that host a resident microbiota: the skin, oral cavity, respiratory tract, intestines, and urogenital tract [8]. In general, there are two main reasons for why a significant portion of the research in the field of pharmacomicrobiomics focuses on the intestinal microbiota. First, approximately 90% of drugs consumed globally are administered orally. Second, the intestinal microbiota is the most diverse of all, consisting of between 30 to 400 trillion microorganisms, and its composition varies based on factors such as ethnicity, dietary intake, and environmental influences [19,20].

Today, especially concerning the intestinal microbiota, it has been demonstrated to play a highly relevant role in how pharmacological treatments are absorbed, distributed, metabolized, excreted, and in their potential toxicity. This mainly occurs through two key mechanisms: drug bioaccumulation and drug metabolism by the microbiota [21].

In the context of pharmacomicrobiomics, the term “drug bioaccumulation” is used to describe the ability of bacteria to store a drug intracellularly without chemically modifying it. This has two consequences: the first is a reduction in drug availability, and the second is due to changes in the composition of the microbial community [22].

Recently, it has been proposed that bioaccumulation is the primary process through which bacteria deplete drugs, even surpassing biotransformation. In a study examining 29 interactions between bacteria and drugs, it was found that 17 of these interactions were related to bioaccumulation, while the remaining 12 were associated with biotransformation [22].

Currently, the mechanisms regulating bioaccumulation by intestinal bacteria are not fully understood. Regarding the accumulation process, some studies have found that drugs such as duloxetine and hydrochlorothiazide have the ability to bind to proteins present in the intestinal microbiota bacteria. Therefore, the binding of drugs to bacterial proteins could be a plausible explanation for accumulation [22,23]. Information about drug transport into bacteria remains limited. However, based on the results of some research, hypotheses can be formulated. For instance, it has been observed that metformin increases the presence of *Akkermansia muciniphila*, a bacterium classified as Gram-negative. These microorganisms possess transport proteins in their outer membrane, such as porins. Among these, the Outer Membrane Protein A (OmpA), due to its nonspecific nature, facilitates the passive transport of many small chemical substances, generally with a molecular weight less than 600 Da, such as metformin [24,25].

Studying the molecular basis related to the transport and accumulation of drugs that are not intended to target intestinal bacteria is an underexplored field that deserves more attention in future research. Research in this field will better enable the understanding of the potential impact of the microbiota on the effectiveness of medical treatments. Additionally, it will help clarify how certain medications can affect the composition and functioning of the intestinal microbiota.

Regarding drug metabolism, the metabolic potential and influence of the microbiota on drug metabolism have been known since 1968 [26]. Intestinal microorganisms can metabolize drugs through processes such as oxidation, reduction, acetylation, deamination, and hydrolysis, among others [27]. One of the most intriguing mechanisms through which gut bacteria metabolize drugs is via CYP enzymes [28,29]. While the human body has a total of 57 identified CYP, bacteria have been found to possess 2979. However, not all bacteria possess these enzymes; for instance, bacteria like *E. coli* lack CYP [30]. Bacterial and archaeal CYP enzymes are soluble and lack membrane-anchoring regions, unlike human enzymes, which are membrane-bound via a transmembrane N-terminal alpha-helical segment. Research on the role of bacterial CYP enzymes extends beyond their involvement in phase I drug metabolism reactions. Experiments have been conducted to modify specific enzymes, such as CYP102A1 (P450 BM3), aiming to alter their structure and potentially affect drug activity. Hence, the study of bacterial CYP enzymes represents a promising research field [31,32]. The differences between drug metabolism and microbial metabolism are illustrated in Figure 2.

Currently, the scientific community is showing great interest in the relationship between drug response and the microbiome or microbiota. This growing attention is reflected in the abundance of publications on this topic. As an example, some of the most recent experimental studies in the field of pharmacomicrobiomics are presented in Table 1.

The table above illustrates the breadth of applications of pharmacomicrobiomics, covering both in vivo and in vitro studies, in animal models and in humans. These studies cover a wide range of conditions, from metabolic disorders to organ transplantation. Advances in next-generation sequencing, metabolomics, transcriptomics, and proteomics suggest that, in the near future, healthcare is likely to experience significant benefits through improved clinical practices, thanks to a better understanding and manipulation of the microbiome for the benefit of the patient [38].

### 2.1. Drug Effect on Microbiome Diversity and Therapeutic Response

The relationship between the microbiome and drugs is two-way. This implies that both the microbiome can influence the response to a drug, and the drug can alter the microbiome, which can result in enhanced—or, in some cases, potentially harmful—therapeutic effects [39].

Although pharmacomicrobiomics was originally conceptualized to describe the metabolizing or storage activity of drugs by the human microbiome, recent studies have broadened the scope of this term to encompass the impact of drugs on the composition of the gut microbiome, as well as its influence on therapeutic response and prognosis in patients. This phenomenon has also been proposed to be called pharmacoecology [10,40,41].

In the medical and scientific community, the effect of medications on the composition of the microbiota is widely recognized, with particular emphasis on the impact of antibiotics. In this context, a well-studied example is the use of cephalosporins, penicillins, clindamycin, and fluoroquinolones, which has been associated with a significant increase in the risk of *Clostridium difficile* infection [42].

While it is understandable that antibiotics have a direct effect on the microbiome due to their antibacterial activity, there are also other drugs that lack this activity but can influence the diversity of the microbiome [43]. A plausible explanation for why drugs not designed to target microorganisms can alter the composition of the microbiome is that these microorganisms tend to accumulate the drug (bioaccumulation), which affects their metabolism, which in turn influences their ability to proliferate [22].

The alteration of the microbiome by medications can have implications for health and the development of diseases. It has been described that the products of metabolism by intestinal microorganisms can influence overall health. An example of this is the ability of antidepressants to modify the composition of the intestinal microbiota. This modification, in turn, can significantly impact the course of the disease due to variations in certain metabolites produced by specific intestinal microorganisms. This situation is particularly interesting in the context of depression, as it provides new insights into how to approach and treat this condition [44,45,46,47,48].

An example is the case of *Candida albicans*, which has the ability to produce acetaldehyde, which is transformed by the enzyme aldehyde dehydrogenase 2 into acetate, a metabolite that can easily cross the blood–brain barrier (BBB) and influence brain neurotransmission. In this context, some antidepressants, such as sertraline, fluoxetine, doxepin, imipramine, and nortriptyline, not only act through their already defined mechanism of action but also inhibit the growth of *Candida albicans* and modify the course of the disease [40,49]. Figure 3 illustrates the effect of antidepressants on *Candida albicans* growth.

Although the previous example with *Candida albicans* simplifies the relationship between medications, microbiota, and therapeutic effect, the interaction between drugs and microorganisms can be significantly more complex. A well-described case involves the effect of metformin on the intestinal microbiota. This drug is widely used to control type 2 diabetes (T2D). The response to metformin differs based on the route of administration, being more effective orally than intravenously. Metformin has a 50% bioavailability, resulting in concentrations 30 to 300 times higher in the jejunum compared to plasma, enabling its interaction with intestinal bacteria. This interaction has been shown to cause changes in the composition of the intestinal microbiota, characterized by an increase in the abundance of *Akkermansia muciniphila* and short-chain fatty acid-producing bacteria following metformin administration. These microbiota changes are associated with improvements in glycemic control in T2D patients [24,50].

A large number of experiments have now shown that different drugs can modify the composition of the microbiota and the response to the drug. Table 2 below shows some of the experimental studies published in 2023, showing the drug and the effect on the microbiota.

### 2.2. Pharmacomicrobiomics Challenges

An interesting publication from 2018 regarding the microbiome mentions, “The field of microbiome research is currently approaching a transition from infancy to toddlerhood. Our means of exploration are changing from crawling to walking, but still require much more growth to acquire fine motor and cognitive skills to make sense of the world around us” [56]. Since 2018, publications using the term “microbiome” have exponentially increased according to PubMed data. Only a small fraction of these publications corresponds to pharmacomicrobiomics.

A well-known proverb states, “When you see your neighbor’s beard on fire, get your water ready”. This could be applied to pharmacomicrobiomics if we use pharmacogenetics as an example. When comparing these two sciences, we find that the term pharmacomicrobiomics was first introduced in 2010. In that same year, according to PubMed data, 1048 articles were published that used the term pharmacogenetics [8]. Today, pharmacogenetics is already applied in clinical practice, as demonstrated by the inclusion of pharmacogenetic information in the labels of over 250 medications by the Food and Drug Administration (FDA). This achievement resulted from numerous studies that spanned from pharmacokinetics; therapeutic drug monitoring; and observational studies, such as cross-sectional studies, case–control studies, and cohort studies, to randomized clinical trials [57].

The challenges facing pharmacomicrobiomics are likely to be greater than those faced by pharmacogenetics before it was incorporated into clinical practice. Unlike the genome, the microbiome can be influenced by factors such as diet, social stress, medications, host genotype, age, lifestyle, habits including smoking and sports, and even urbanization [58].

Finally, it is crucial to consider that various meta-omic techniques can be employed in pharmacomicrobiomics. The most common technique in studying the intestinal microbiome is metagenomics, which focuses on analyzing the genetic material of the microbial community residing in the digestive tract. However, metagenomics has the limitation of exclusively focusing on genes, which hinders a complete understanding of temporal dynamics and functional activities of microbial populations [59].

To complement metagenomics, other techniques such as metatranscriptomics and metaproteomics have been introduced. These allow for the study of interactions between bacterial communities and the host at the gene expression level, considering both transcription and proteins, respectively [59]. Figure 4 presents a chart proposing the integration of various omic techniques that could be used in pharmacomicrobiomics.

## 3. Effect of an Infection on Drug Response

### 3.1. Difference between Pharmacomicrobiomics and Drug–Infection Interaction

There are two fundamental differences between pharmacomicrobiomics and drug–infection interaction. The first one lies in the type of microorganism that influences the drug response. In pharmacomicrobiomics, the microbiome is the protagonist, whereas in drug–infection interaction, it involves a pathogenic microorganism [20,41,60].

Pathogenic microorganisms are those capable of causing diseases, as they are transmissible and, in some cases, have developed the ability to evade cellular defenses. Only a small percentage of microbes are inherently pathogenic. Pathogenic microorganisms include some viruses, bacteria, prions, fungi, protozoa, and parasites [60].

The second difference is that, unlike pharmacomicrobiomics, which modifies the drug response through bioaccumulation or metabolism, infections can alter the drug response mainly through inflammation and the regulation of CYP enzymes [21,61,62].

### 3.2. Inflammation as a Result of Infection Modifies Drug Response

Inflammation is a response to aggression, whether of endogenous or exogenous origin, and can manifest acutely or chronically. It plays a prominent role in numerous diseases, including infections [63]. While inflammation is a complex and highly coordinated process involving multiple cell types and molecules operating in a cascading network, cytokines play a particularly relevant role in this process [64]. For several years, it has been proven that inflammation has a significant impact on drug metabolism. This is partly because elevated levels of proinflammatory cytokines lead to a negative regulation of CYP enzymes, which play a fundamental role in drug metabolism [61,65,66].

CYP enzymes are polymorphic proteins associated with a heme molecule and are capable of absorbing light at a wavelength of approximately 450 nm when exposed to carbon monoxide. These enzymes play an essential role in the biosynthesis of compounds such as steroids, prostacyclin, and thromboxane A2. While CYP enzymes are found in a wide variety of tissues, their expression is most prominent in the liver and small intestine. Regarding drug metabolism, a specific group of CYP enzymes, including CYP 1A2, 2B6, 2D6, 2C8, 2C9, 2C19, and 3A4, are responsible for metabolizing most drugs [5,61].

CYP enzymes during inflammation can be repressed by different mechanisms. These include the transcriptional downregulation of transcription factors, interference with nuclear transcription factor dimerization and translocation, alteration of C/EBP-enriched signaling in the liver, direct regulation by NF-κB, and various post-transcriptional mechanisms [66]. It has recently been proposed that the reduction in CYP enzyme activity during inflammation, in the context of an infection, is due to a physiological response. This response involves a shift from a metabolic mode to a defensive mode, allowing the cell to concentrate its resources on fighting the infection [67].

Regardless of the physiological cause, infections impact CYP enzyme activity due to the inflammatory process they trigger [63,68,69]. A prominent example of this is the disease caused by SARS-CoV-2, known as COVID-19. This disease follows a progression divided into three stages: the viral invasion phase, the pulmonary immunoinflammatory phase, and the hyperinflammatory phase. Inflammation is its distinguishing feature, marked by increased NF-κB signaling, which in turn induces the production of proinflammatory cytokines such as IL-6, IL-2, TNF-α, and IFN-γ [70].

In this context, it has been documented that the increased proinflammatory cytokines induced by COVID-19 impact drug metabolism as they interfere with the regulation of CYP enzymes and drug transporter expression [68,71,72]. In humans, it has been observed that inflammation caused by COVID-19 reduces CYP3A activity, which, in turn, affects the metabolism of midazolam. Furthermore, two independent studies consistently reported abnormally elevated levels of lopinavir and ritonavir in COVID-19 patients, suggesting that this could be due to the negative regulation of CYP3A [73,74,75].

Regarding COVID-19, the response to treatment is not solely related to the disease itself. In 2021 and 2022, two cases were reported, in which increased levels and adverse effects of clozapine were observed in patients who had been vaccinated against COVID-19. These specific vaccines were Moderna’s Spikevax and Pfizer-BioNTech’s vaccine. In both cases, the adverse reaction was associated with inflammation and CYP1A2 activity. It is important to note that this adverse reaction was short lived [76,77].

In the case of the human immunodeficiency virus (HIV), it has been described to affect CYP enzyme levels. People infected with HIV have shown a reduction in hepatic CYP3A4 and CYP2D6 enzyme activity compared to uninfected individuals [78,79]. However, the interpretation of these findings is not entirely conclusive, as other research has not found changes in drug metabolism in HIV patients, and, in other studies, an increase in CYP3A4 expression has even been observed in HIV patients receiving antiretroviral therapy [80,81].

For HIV patients, future research aimed at determining the impact of infection-generated cytokines on the pharmacokinetics of antiretroviral drugs should address various variables. This includes individual gene expression, the possible co-infection of HIV with hepatitis B, the presence of liver disease, the anti-inflammatory effects of drug therapy, and study design, among other factors. A consideration of these elements is crucial for obtaining clearer and more accurate results in this area of research [61,80].

As evidence continues to accumulate, it is likely that closer medical monitoring may be needed in the future for patients with an infection who are also being treated with drugs metabolized by CYP enzymes to prevent possible overdoses and toxicity.

### 3.3. Other Mechanisms by Which Infections May Affect Drug Response

#### 3.3.1. Alterations in Gastrointestinal Motility and Drug Absorption

It has been proposed that gastrointestinal infections may affect the availability of certain drugs due to various factors, such as changes in intestinal transit speed or the pH of gastrointestinal fluids; however, information is limited [82,83]. In the context of alterations in intestinal transit caused by an infection, it is important to consider infectious diarrhea. While this condition can influence the absorption of a medication, its specific impact can vary considerably, depending on several factors, including the severity of the diarrhea, its duration, the overall health of the individual, and the underlying infectious agent, as infectious diarrhea can be caused by viral, bacterial, or parasitic infections [84].

Overall, the evidence supporting the impact of an infection that causes diarrhea on drug absorption is limited. As a result of this research, very few studies directly addressed the impact of infectious diarrhea on drug absorption. In one study, it was investigated whether the absorption of proguanil and chloroquine, used for malaria prevention, was affected by traveler’s diarrhea. The results indicated that patients with traveler’s diarrhea had significantly lower maximum concentrations and absorption coefficients for proguanil compared to subjects without diarrhea [85]. In HIV patients, it has been observed that diarrhea reduces the absorption of tuberculosis drugs [86,87]. Another study evaluated the bioavailability of ciprofloxacin in patients with infectious diarrhea and concluded that the drug was well absorbed in patients with acute diarrhea, with adequate blood levels despite the presence of diarrhea [88].

The lack of research directly addressing the mechanisms by which infectious diarrhea affects drug absorption is largely due to the fact that most patients with acute diarrhea typically present mild and transient symptoms. Additionally, in severe cases of diarrhea, the priority is to immediately address the clinical condition rather than evaluating whether diarrhea might modify the absorption of a prescribed medication [89]. Further research is needed to fully understand the impact of infectious diarrhea on drug absorption and its implications in clinical practice.

#### 3.3.2. Pharmacological Effect Mimicry

A poorly described mechanism by which an infection might modify the response to a drug is through mimicry of the drug effect. An example of this is human adenovirus 36 (HAdV-36), which has been associated with obesity and changes in glucose and lipid metabolism, with long-term effects, such as the irreversible expansion of adipose tissue, even after the resolution of the acute phase of infection [90]. HAdV-36 increases peroxisome proliferator-activated receptor-γ (PPAR-γ) expression in the same way as thiazolidinediones, which are used to increase insulin sensitization in patients with type 2 diabetes mellitus [91,92,93]. This virus could potentially influence the response to drugs used in lipid or glucose control, although there is no evidence so far that HAdV-36 modifies the response to drugs such as metformin [94]. To our knowledge, there is no other virus that can mimic the effect of a drug.

#### 3.3.3. Unknown Mechanisms: The Case of *Helicobacter pylori* and Levodopa

*Helicobacter pylori* (HP) has been documented to cause inflammation at the intestinal level, delay gastric emptying, and possibly affect the absorption of drugs, such as Levodopa [95]. In this context, it has been observed that people with Parkinson’s disease who have HP infection show a poor response to Levodopa and experience increased severity of motor symptoms [96]. However, the elimination of HP has been reported to improve tremor, although it does not change the bioavailability of the drug [97]. All this information underscores the need for further research to understand how HP influences drug interactions with drugs, such as Levodopa, and other potential drugs.

## 4. Materials and Methods

This review was carried out with the aim of exploring the influence of the microbiome and infections on the response to a drug. To this end, an in-depth search of the scientific literature was conducted in the PubMed and Web of Science databases. A specific search strategy was designed that included a series of key terms and Boolean operators. The key terms used in the search strategy were: “Pharmacomicrobiomics”, “microbiome”, “microbiota”, “infection”, “medication response”, “impact medication effectiveness”, “influence drug response”, “alter medication outcome”, “affect drug efficacy”, “change medication reaction”, “modify drug response”, “influence medication effectiveness”, and “impact drug outcome”. A combination of “AND” and “OR” operators was used to ensure the inclusion of relevant studies addressing pharmacomicrobiomics and the relationship between infections and medication response. We searched for articles published up to the cut-off date of this review (July 2023), restricting the search for articles to those published in English or Spanish.

To achieve an even more precise search, we turned to the EvidenceHunt Chat (available on the http://evidencehunt.com website (accessed on 21 August 2023)), an artificial intelligence-powered tool for searching clinical evidence. In this process, we formulated specific questions related to the various topics addressed in this article and then reviewed the articles upon which EvidenceHunt based its answers.

The selection of articles was initially based on the relevance of titles and abstracts, excluding those that did not align with the aim of this review. Only studies directly addressing the term “pharmacomicrobiomics” or specifically discussing the impact of infections caused by bacteria, viruses, or other microorganisms on the response to any drug were included in this review. The information obtained from the selected studies was analyzed and synthesized to provide an overview of the relationship between infections and medication response.

## 5. Conclusions

Humans have maintained an intimate relationship with the microbial world throughout history. Many microorganisms work in symbiosis and are essential for human health and well-being. The study of this relationship between microorganisms and humans has generated great interest in the scientific community. Recently, it has been documented that microorganisms have the ability to influence the response to drug therapy.

Pharmacomicrobiomics is an emerging field that seeks to understand how variations in the microbiome affect the distribution, action, and potential toxicity of drugs. Bacteria within the microbiome primarily interact with drugs through two processes: bioaccumulation and biotransformation. Bioaccumulation refers to the bacteria’s ability to store drugs within their cells, which can reduce drug availability and alter the microbial community composition; this phenomenon has been proposed as “pharmacoecology”. On the other hand, biotransformation describes how the microbiome metabolizes drugs, producing metabolites that can influence drug responses, including the formation of toxic metabolites, known as “toxicomicrobiomics”.

Although pharmacomicrobiomics focuses on the microbiome, it does not include pathogenic microorganisms, which, unlike the microbiome, can regulate the host’s CYP enzyme expression and modify drug responses. We propose introducing the term “pharmaco-infection” to describe the influence of pathogenic microorganisms on drug responses. With this proposal, we seek to identify and unify the disciplines that contribute to the study of the interaction between microorganisms and the response to drugs by the host. This will not only allow for better organization and understanding of the knowledge in this field, but will also promote specialization, promote the adoption of a common language and specific terminology, and facilitate the evaluation of research progress.

## Figures and Tables

**Figure 1 ijms-24-17100-f001:**
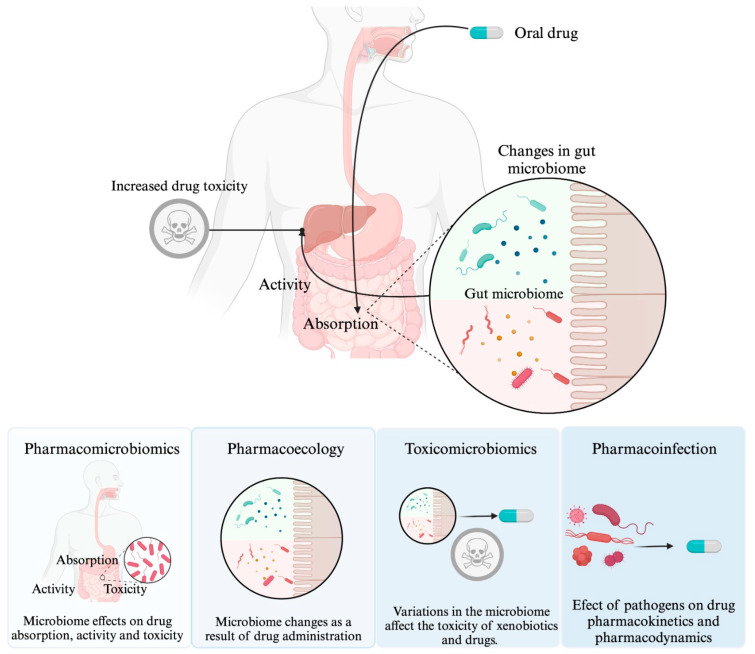
Effect of microorganisms on the host response to drugs. This depiction illustrates the distinction between four key concepts: pharmacomicrobiomics, toxicomicrobiomics, pharmacoecology, and drug–infection interaction. Pharmacomicrobiomics focuses on how variations in the microbiome affect drug disposition, action, and toxicity. Toxicomicrobiomics, meanwhile, explores the influence of the microbiome on drug metabolism and toxicity. Meanwhile, pharmacoecology focuses on the modifications in the microbiome that result from drug administration. Finally, drug–infection interaction investigates the impact of pathogenic microorganisms on drug response.

**Figure 2 ijms-24-17100-f002:**
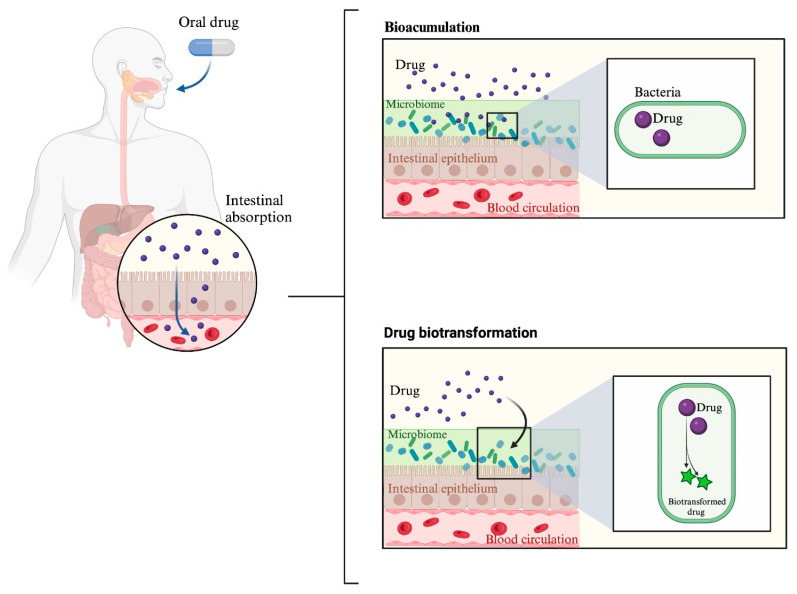
Effects of the microbiome on drug response. In this figure, the processes of drug bioaccumulation and metabolism are depicted. Bioaccumulation refers to the ability of bacteria to store certain drugs intracellularly. Drug metabolism, on the other hand, indicates how the microbiome participates in the metabolization of drugs. The arrow in the box corresponding to ‘drug biotransformation’ indicates that bacteria interact with the drug to metabolize it.

**Figure 3 ijms-24-17100-f003:**
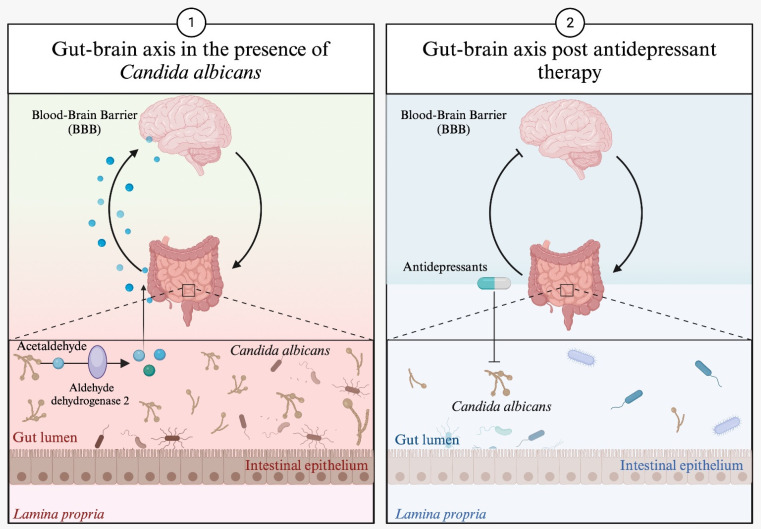
Effect of antidepressants on *Candida albicans*. In the section labelled “1” in Figure 3, we can see how *Candida albicans*, through its metabolic activity, can influence brain neurotransmission. In section “2” of Figure 3, it is shown how antidepressants inhibit the growth of *Candida albicans*, leading to a reduction in the production of its metabolic products and, as a result, modifying the course of depression.

**Figure 4 ijms-24-17100-f004:**
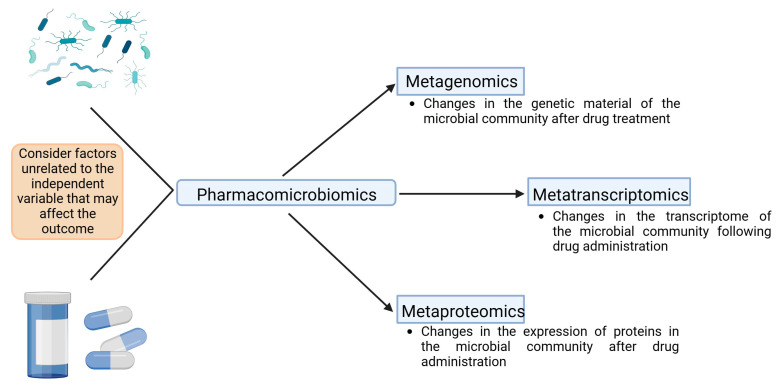
Multiple levels of meta-omics that can be used in pharmacomicrobiomics studies. In pharmacomicrobiomics, various meta-omic techniques can be applied. Firstly, metagenomics is crucial for identifying the microbial genome. This technique can be complemented with metatranscriptomics and metaproteomics, enabling the understanding of expression levels and the proteins that the microbiome produces in the presence of drugs. An analysis of protein expression provides insights into the proteins used by microorganisms to metabolize, transport, and accumulate drugs. Additionally, it helps clarify how the microbiota interacts with its environment, including other microorganisms. The image does not overlook factors that could influence the independent variable, which, in this case, corresponds to drug administration. In pharmacomicrobiomic studies, monitoring and controlling variables that could alter the microbiota are essential. This allows for a more precise attribution of any changes in the microbial community to the effect of the drug.

**Table 1 ijms-24-17100-t001:** Experimental studies in pharmacomicrobiomics across various models. The table presents the most recent studies in pharmacomicrobiomics, encompassing investigations in humans, mice, and in vitro. These studies examine the impact of the microbiome on drugs, either in terms of drug bioaccumulation, metabolism, or both.

Drug Involved	Affected Process	Associated Microorganisms	Clinical Effect	Condition for Which Studied	Organism in Which the Study Was Conducted
Chemotherapy and/or immunotherapy	Not described	*L. mucosae* and *L. salivarius*	Favorable response	Metastatic/unresectable HER2-negative gastric/gastroesophageal junction adenocarcinoma	Humans [33]
FOLFOX regimen	Drug metabolism	*Akkermansia muciniphila*	Better therapeutic effect	Colon Cancer	Mice [34]
Hydrochlorothiazide	Bioaccumulation	Gram-negative enterobacteriaceae	Impair glucose tolerance	Metabolic control	Mice [23]
Mycophenolate mofetil	Drug metabolism	Bacteroides vulgatus, Bacteroides stercoris and Bacteroides thetaiotaomicron	Graft-versus-host disease risk reduction	Transplantation	Humans [35]
Simvastatin	Drug metabolism and bioaccumulation	Probiotic bacteria	Alteration of simvastatin bioavailability and therapeutic effect	Metabolic control	in vitro [36]
Statins	Drug metabolism	Bacteroides	Intense statin responses	Metabolic control	Humans [37]

**Table 2 ijms-24-17100-t002:** Impact of different drugs on intestinal microbiota diversity. This table presents experimental studies on different non-antimicrobial medications that affect the composition of intestinal microbiota. Unlike Table 1, specific processes affected are not detailed here, as all cases show an impact on microbiota composition. * A herbal product used mainly in traditional or alternative medicine.

Drug Involved	Condition for Which Studied	Microbiome Effects	Clinical Effect	Organism in Which the Study Was Conducted
5-aminosalicylic acid	Inflammatory bowel disease	Higher abundance of Firmicutes and lower abundance of Bacteroidetes	Not affecting intestinal morphology	Mice [51]
Dapagliflozin	Heart failure	Decreased the ratio of Firmicutes/Bacteroidetes	Reduction in inflammation, infarct area, and cardiac fibrosis	Mice [52]
Lactulose	Acute pancreatitis	Bifidobacterium enriched	Reduction in serum levels of proinflammatory cytokines and intestinal permeability index.	Humans [53]
Roxadustat	Renal anemia	Increase in short-chain fatty acid-producing bacteria	Relief of renal anemia	Humans [54]
Sinomenine *	Arthritis	Increase in *Lactobacillus* spp.	Relief of arthritis symptoms	Rat [55]

## Data Availability

Data sharing not applicable. No new data were created or analyzed in this study. Data sharing is not applicable to this article.

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
