# Peer review of "Pharmacomicrobiomics and Drug–Infection Interactions: The Impact of Commensal, Symbiotic and Pathogenic Microorganisms on a Host Response to Drug Therapy"

_ijms, 2023, doi:10.3390/ijms242317100_

Round 1

Reviewer 1 Report

Comments and Suggestions for Authors

This review manuscript by Torres-Carrillo merits consideration for publication in IJMS. However, the authors should give a glimpse, at least, in the bacterial arsenal that metabolizes and modifies drugs. For example, they do not mention that the microbiome, most notably the bacteria that comprise it, could also possess CYP enzymes (however, E. coli does not possess any CYPs). Also, they do not mention what potential bacterial membrane proteins, i.e., transporters, could lead to the bio-accumulation of certain drugs. For example the outer membrane porins in Gram-negative bacteria, such as E. coli, allow transit of polar compounds less than 800 Da. It would be interesting to comment on this plausible mechanism for bioaccumulation, as this process seems to be important in drug regimens, as they carefully state. In addition, lipophilic drugs cross directly through bacterial membranes, if there is a disruption in the bilayer. Does this mean that bacteria will alter permeability of their membranes to accumulate certain types of drugs? They should comment on whether this is active or passive process, to the best of their knowledge. They should also that a transformed product from one bacterium can act as a substrate for another bacterium, hence the definition of a holomicrobiome; is this the same fate for drugs? If possible, please include the anti-diabetic drug metformin, in the case of a drug affecting the microbiome, leading to a therapeutic effect on the host, which has been confirmed. These points would greatly enhance their review. I like their implementation of novel terms in their review, but the molecular basis, such as the ones mentioned above, of this terms should be discussed more deeply. The clinical cases of microbiome-drug interaction they report are valuable, especially the COVID-19 effect on CYP enzymes.

Minor points, Tables 1, 2 have their legends before the actual table, please correct. 

Please italicize Candida albicans and Helicobacter pylori throughout the text. 

Reviewer 2 Report

Comments and Suggestions for Authors

In the manuscript submitted to me for review entitled "Pharmacomicrobiomics and drug-infection interaction: Impact of commensal, symbiotic and pathogenic microorganisms on host response to drug therapy“ the authors Norma Torres-Carrillo, Erika Martínez-López, Nora Magdalena Torres-Carrillo, Andres López-Quintero, José Miguel Moreno-Ortiz, Anahí González-Mercado and Itzae Adonai Gutiérrez-Hurtado conduct a comprehensive review analyzing the knowledge about the importance of microorganisms in the host response to drugs - how the microbiome affects drug metabolism and toxicity and the modifications that occur in the microbiome as a result of drug administration.

The authors summarize the information presented using 3 well-designed figures and 2 tables. In support of their study, the authors used 88 literature sources that cover information from the last 6 decades. The main information is from the last 5 years - 62 references (nearly 3/4 of the total number), of which 22 (1/4 of the total number) are from the current year 2023.

My remarks and recommendations to the authors are:

1.     References [3, 5] are cited on line 51, but the citation of reference #4 is missing both here and in a previous position in the manuscript. I assume it is the result of a typo and should be spelled [3 - 5]. Let it be corrected, or add reference #4 if it is omitted.

2.     In the Author Contributions section at the end of the manuscript, each of the authors must indicate what contribution they have to the realization of the manuscript.

3.     In the section Conflicts of Interest: at the end of the manuscript it must be stated that the authors do not have any conflict - a mandatory condition for publication by IJMS.

4.     Some of the references do not list all the authors (Nos. 20, 22, 25, 28, 30, 36, 43, 65, 70, 77, 84 and 86). Personally, when I read an article, I prefer the references to be fully written, instead of having to search for some of the authors. I think it would be helpful to your readers if all authors in all references are listed.

Please find additional comments in the attached file

Reviewer 3 Report

Comments and Suggestions for Authors

This review has investigated the newer field of Omics Sciences, which is growing rapidly. This review is written nicely, and I recommend this to be published after minor revision.

1. What are the characteristics metabolic pathways affected during this interaction? Can be mentioned in the conclusion.

2. Are the listed drugs clinically approved?

3. There are several terms in Omics, how the experimental model can be designed for carrying out pharmacomicrobiomics? Provide a graph for experimental set-up for this Omics workflow.

Round 2

Reviewer 1 Report

Comments and Suggestions for Authors

Thank you for addressing my concerns. This is an ever evolving field, and I think your review sums it up quite nicely.